# FlowFeat: Pixel-Dense Embedding of Motion Profiles

**Nikita Araslanov**[1,2]    **Anna Sonnweber**[1]    **Daniel Cremers**[1,2]

[1]TU Munich    [2]MCML

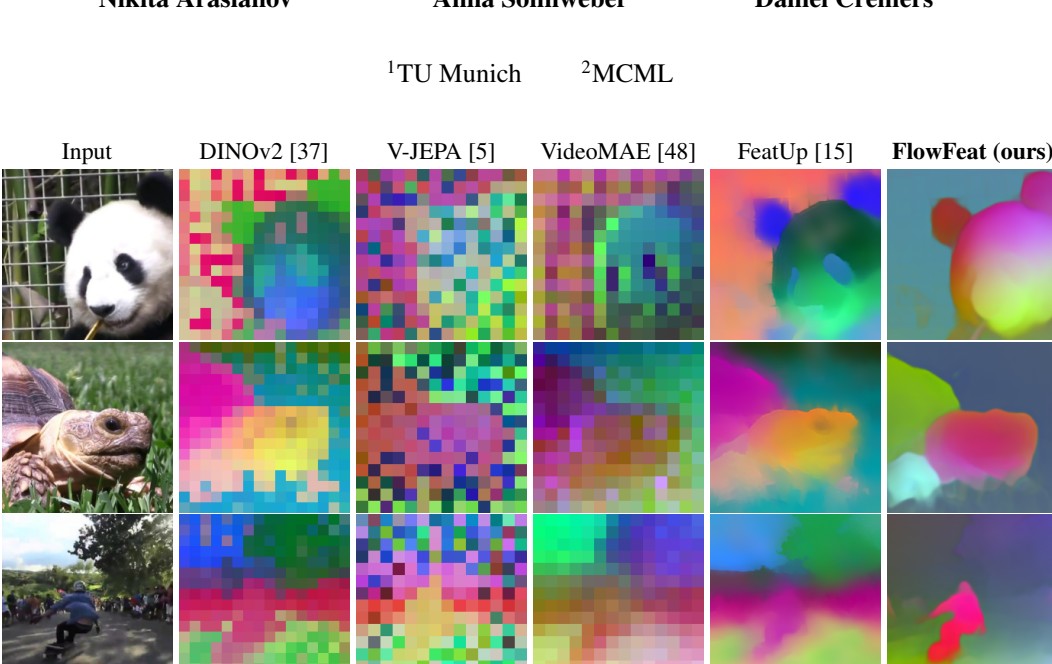

Figure 1: **FlowFeat** is a versatile feature representation at pixel-level resolution. Embedding profiles of plausible motion, FlowFeat stands out from existing techniques by offering excellent spatial precision coupled with temporal consistency. Here, we visualise (using PCA with three principal components) a comparison of FlowFeat with the feature maps of the state-of-the-art vision encoders.

## Abstract

Dense and versatile image representations underpin the success of virtually all computer vision applications. However, state-of-the-art networks, such as transformers, produce low-resolution feature grids, which are suboptimal for dense prediction tasks. To address this limitation, we present FlowFeat, a high-resolution and multi-task feature representation. The key ingredient behind FlowFeat is a novel distillation technique that embeds a distribution of plausible apparent motions, or *motion profiles*. By leveraging optical flow networks and diverse video data, we develop an effective self-supervised training framework that statistically approximates the apparent motion. With its remarkable level of spatial detail, FlowFeat encodes a compelling degree of geometric and semantic cues while exhibiting high temporal consistency. Empirically, FlowFeat significantly enhances the representational power of five state-of-the-art encoders and alternative upsampling strategies across three dense tasks: video object segmentation, monocular depth estimation and semantic segmentation. Training FlowFeat is computationally inexpensive and robust to inaccurate flow estimation, remaining highly effective even when using unsupervised flow networks. Our work takes a step forward towards reliable and versatile dense image representations.

Project website: https://tum-vision.github.io/flowfeat.
Code and pre-trained models (Apache-2.0 License): https://github.com/tum-vision/flowfeat.

39th Conference on Neural Information Processing Systems (NeurIPS 2025).

# 1 Introduction

The feature maps of state-of-the-art self-supervised encoders (*e.g.* [9, 18]) have drastically downsampled spatial resolutions (*e.g.* by a factor of 16), as illustrated in Fig. 1. While such downsampling improves the computational efficiency of deep networks, it compromises on the accuracy of dense prediction tasks, where spatial detail is crucial. Upsampling techniques, such as those based on bilateral filters [15], can recover feature detail to an impressive degree. However, bilateral upsampling incurs a tangible computational cost and struggles under challenging illumination scenarios (*cf*. Fig. 1, third row). Alternatively, one could equip encoders with a lightweight decoder module, such as DPT [41]. However, training such decoders without human annotation is highly non-trivial. Building on this motivation, we present *FlowFeat*, a multi-task pixel-level image representation obtained in a label-efficient (or even label-free) manner.

Different from much of the existing work on representation learning, FlowFeat derives from dense motion patterns rather than the static appearance alone [3, 10, 18, 37]. While FlowFeat is a monocular model operating on a *single* input image at test time, it uses unlabelled videos for training to embed motion patterns into a pixel-level representation. Motion patterns are foundational to visual perception [33]; they encode the compositional nature of visual scenes, encompassing both semantically and geometrically meaningful phenomena. However, as Fig. 1 illustrates, video-based learning still fails to provide representations that are dense, versatile and effective [5, 13, 21].

As a step forward, we synergise state-of-the-art optical flow networks and real-world video data. On the one hand, modern optical flow networks produce dense motion estimates with outstanding accuracy, even in challenging settings [45, 47, 52]. On the other hand, datasets of casual videos provide a treasure trove of motion and scene diversity [23, 54]. Combining both ingredients in a joint learning framework, FlowFeat requires no manual annotation. Optical flow networks train predominantly on synthetically generated labels or even with self-supervision [35, 44]; video datasets derive from real-world benchmarks and require minimal curation (*e.g.* montage filtering). The key technical challenge is distilling the apparent motion in a fashion accommodating its stochastic nature.

FlowFeat addresses this challenge with a simple idea. We estimate the feature representation with a *distribution* of linear transformations. Intuitively, for a given image and a flow estimate w.r.t. a randomly sampled counterpart, FlowFeat is trained to admit a linear transformation approximating the flow. Specifically, every training iteration estimates a *lower bound* of this transformation *on-the-fly* using a least-squares formulation. The statistical nature of this lower-bound approximation (due to sampling of the image pair) accommodates motion stochasticity and proves crucial for dealing with inaccurate flow and occasional static scenes. Consequently, the distribution of linear transformations allows FlowFeat to embed a distribution of plausible motion, or *motion profiles* [42].

Overall, our work presents two contributions. First, we develop an effective self-supervised training framework that exploits the synergetic power of flow networks and large video datasets to embed motion profiles. Our framework is efficient at training time and can run comfortably within academic infrastructures. Second, we extensively evaluate the learned representation, FlowFeat, on three diverse tasks of dense prediction: video object segmentation (VOS), monocular depth estimation and semantic segmentation. Our analyses reveal a consistent benefit of FlowFeat across all tasks, exhibiting a compelling degree of temporal consistency and spatial detail. Furthermore, FlowFeat has appealing practical properties: *(i)* it is runtime- and label-efficient; *(ii)* it scales well with varying input resolution without the need for model fine-tuning, and *(iii)* it facilitates simple post-processing tasks, enhancing the quality of dense predictions without additional training.

# 2 Related Work

A substantial effort towards unsupervised feature representations has focused on learning from large image sets [3, 11, 16]. This development spans multiple axes of pursuit, such as model efficiency [9, 55], scalability [18, 37] and framework architecture [12]. Although pre-training from image sets dominates the research landscape in unsupervised learning, there have been natural extensions of image-based frameworks to learning from video data [5, 13, 48]. However, it remains challenging to obtain *spatio-temporal* representations that are both dense (*i.e.* pixel-level) and temporally consistent [2, 10, 51]. Central to learning spatio-temporal representations is the design of the pre-text task. One prominent technique is *cycle consistency* [21, 27, 43, 51]. It constructs a temporal palindrome

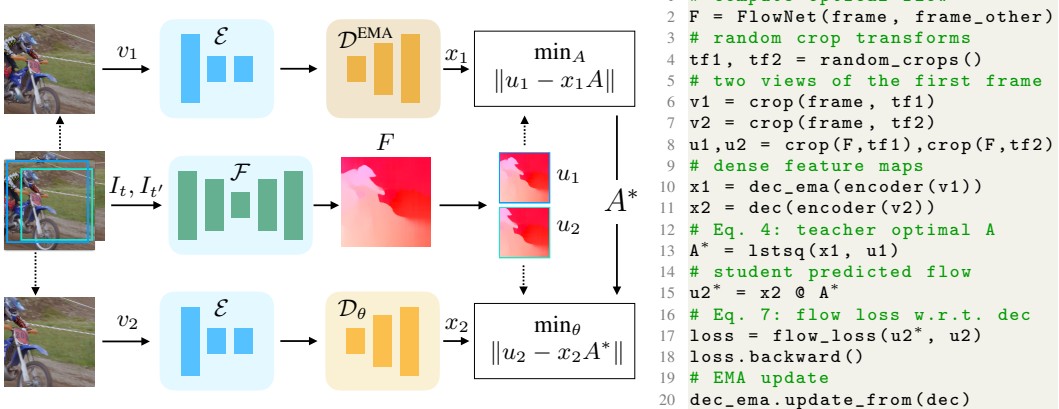

```
1  # compute optical flow
2  F = FlowNet(frame, frame_other)
3  # random crop transforms
4  tf1, tf2 = random_crops()
5  # two views of the first frame
6  v1 = crop(frame, tf1)
7  v2 = crop(frame, tf2)
8  u1,u2 = crop(F,tf1),crop(F,tf2)
9  # dense feature maps
10 x1 = dec_ema(encoder(v1))
11 x2 = dec(encoder(v2))
12 # Eq. 4: teacher optimal A
13 A* = lstsq(x1, u1)
14 # student predicted flow
15 u2* = x2 @ A*
16 # Eq. 7: flow loss w.r.t. dec
17 loss = flow_loss(u2*, u2)
18 loss.backward()
19 # EMA update
20 dec_ema.update_from(dec)
```

Figure 2: **Embedding motion profiles**: FlowFeat relies on the exponentially moving average (EMA) teacher model and learns to reconstruct apparent motion with a distribution of linear transformations. For a given frame $I_t$, we randomly sample its temporal counterpart $I_{t'}$. A pre-trained network $\mathcal{F}$ computes optical flow $F_{(t \to t')}$. We generate two overlapping random crops of frame $I_t$ and feed the resulting views $v_1$ and $v_2$ to the teacher and the student networks, respectively. Obtaining the optimal linear transform $A^*$ on-the-fly with ridge regression in the teacher branch, we compute the reconstruction loss w.r.t. the flow crop $u_2$ to update the student parameters $\theta$ with gradient descent.

from a video sequence, ensuring consistency of a putative state in a forward and backward directions. Contrastive learning underpins another broad category of the research effort [39]. The main ideas are: constructing a reliable set of positive and negative samples [22]; combining learning on pixel, frame, and video levels [50, 53]; or jointly representing a video clip with a limited set of contrastive anchors [2]. Unlike these feature-based techniques, which have limited resolution, photometric-based learning, such as colourisation, relies on natural radiance-based appearance [49]. Lai et al. [28] leverage this technique in video-based learning, reconstructing the target frame from previous frames observed in the CIELAB colour space.

Feature upsampling strategies, such as FeatUp [15] and LoftUp [20] are closely related to our work. In contrast to bilateral upsampling [15, 26], FlowFeat is more computationally efficient and has *complementary* properties to the low-resolution encoder features. Unlike contemporaneous work [20] leveraging SAM [25], FlowFeat is label-efficient and can be trained in an unsupervised manner.

Representation learning by or with motion estimation is not new [17, 32, 38] and traces back to the earlier works on trajectory clustering and motion-based segmentation [7, 14, 29, 56]. Training FlowFeat is efficient, since it does not require pairwise sampling [32]; nor does it require object discovery [19, 38]. Instead, FlowFeat learns directly from optical flow provided from off-the-shelf networks with a distribution of linear transformations. This approach takes primary inspiration from motion profiles, which model a distribution of velocities at a given pixel [42]. *Embedding* motion profiles, FlowFeat enhances downstream accuracy of the baseline representation across diverse tasks.

## 3  Embedding Motion Profiles

**Linear maps for optical flow.**   To obtain pixel-level features enhancing the low-resolution representation of pre-trained encoders, we estimate apparent motion in real-world video sequences. Off-the-shelf optical flow models exhibit exceptional generalisation, despite being trained on synthetic scenes [47, 52] or even with self-supervision [44]. However, distilling motion estimates into a *monocular* model (in contrast to previous work [30]), is highly non-trivial due to motion stochasticity.[1] Overcoming this issue, we train an image representation $\mathcal{H}_\theta(I) = x$ such that for *any* temporally neighbouring frame of $I$, there exists a linear operator on $x$ which approximates the optical flow w.r.t. that neighbour. Since we estimate the linear operator uniquely for each frame neighbour, the learned

---

[1] Naïvely approximating optical flow with a single linear layer unsurprisingly fails, as we verify in Sec. 4.4.

representation $x$ would embed *statistical motion patterns* for each input image $I$ – an idea inspired by motion profiles [42].

Given image $I_t$ and its temporal neighbour $I_{t'}$ of resolution $H \times W (=: N)$, we formulate the idea above with the following flow reconstruction objective (where $\| \cdot \|$ denotes an "entry-wise" norm):

$$\min_{\theta, A} \quad \mathbb{E}_{I_t, I_{t'}} \left[ \| \mathcal{F}(I_t, I_{t'}) - \mathcal{H}_\theta(I_t)A \| \right], \tag{1}$$

where $\mathcal{F}(I_t, I_{t'}) \in \mathbb{R}^{N \times 2}$ is the optical flow from a pre-trained network [47, 52]; $\mathcal{H}_\theta(I_t) \in \mathbb{R}^{N \times d}$ is our learned pixel-level feature representation and $A \in \mathbb{R}^{d \times 2}$ is a linear operator. Note that since $\mathcal{H}_\theta$ and $A$ are both unknown, Eq. (1) is an ill-posed problem due to scale ambiguity.[2] Therefore, we propose to compute the corresponding loss in two steps: *(i)* computing a lower-bound $A^*$ with a surrogate teacher network, while keeping $\mathcal{H}$ fixed; *(ii)* computing the gradient w.r.t. $\theta$ of the original network by swapping $A^*$ into Eq. (1) as the lower-bound linear approximation.

**Student-teacher framework.**  Fig. 2 illustrates the framework and the corresponding training algorithm. Leveraging the mean teacher as the training model [46], our network $\mathcal{H}_\theta := \mathcal{D}_\theta \circ \mathcal{E}$ comprises a fixed (pre-trained) encoder $\mathcal{E}$ and a trained lightweight decoder $\mathcal{D}_\theta$, which outputs a dense feature representation of dimensionality $d$. The teacher model $\mathcal{H}^{\text{EMA}}$ is equivalent to $\mathcal{H}_\theta$ with the exception of the decoder $\mathcal{D}^{\text{EMA}}$, which is an exponential moving average of $\mathcal{D}_\theta$.

To construct the training batch, we sample two frames, $I_t$ and $I_{t'}$, where $I_{t'}$ could be selected from a temporal window around $I_t$. We first compute optical flow $\mathcal{F}(I_t, I_{t'}) \in \mathbb{R}^{N \times 2}$ with a network pre-trained on synthetic data [47, 52] or with self-supervision [44]. Generating two overlapping random crops of the first frame $I_t$, we feed the corresponding views $v_1$ and $v_2$ to the teacher and student models, respectively. Using the teacher output, we solve a least-squares problem:

$$A^* = \text{argmin}_A \| u_1 - \mathcal{H}^{\text{EMA}}(v_1)A \|_2, \tag{2}$$

where $u_1$ is the crop of the optical flow corresponding to view $v_1$. In practice, we solve Eq. (2) with ridge regression, which yields stable solutions in the presence of inaccurate flow estimates and improves training stability (*cf*. Sec. 4.4 for empirical results). Specifically, we solve

$$\min_A \| u_1 - \mathcal{H}^{\text{EMA}}(v_1)A \|_2 + \gamma \|A\|_2, \tag{3}$$

in each training iteration. Here, $\gamma$ is a ridge hyperparameter fixed for all models. Setting $x_1 := \mathcal{H}^{\text{EMA}}(v_1)$ to simplify the notation, the closed-form solution of Eq. (3) is naturally

$$A^* = (x_1^T x_1 + \gamma I)^{-1} x_1^T u_1. \tag{4}$$

Note that the first term has the *feature* dimensions, $d \times d$, fixed to $d = 128$ in our experiments. Therefore, computing Eq. (4) has a negligible computational cost. In contrast to previous work [32], our framework remains computationally efficient regardless of the image resolution.

Fixing $A^*$, we now formulate the flow reconstruction loss w.r.t. the student parameters of $\mathcal{H}_\theta$ as

$$\mathcal{L}_{L1}(u_2, v_2) = \| u_2 - \mathcal{H}_\theta(v_2)A^* \|_1. \tag{5}$$

The loss encourages the two overlapping crops of an input frame to admit the *same* linear mapping $A^*$ from the features to optical flow, thereby promoting grouping of pixels with similar motion patterns. Note that for zero motion (*i.e.* static scenes) the solution is $A^* = 0$, which yields zero gradient for the reconstruction term, effectively discarding such training samples in the learning process. As we also verify in the ablation study (*cf*. Tab. 3), ridge regularization and the robust $L_1$ loss improve resilience of the framework to inaccuracies in the estimated target flow $u_1$ and $u_2$, respectively.

**Focal gradient matching.**  Motion boundaries in optical flow are well-known to reveal semantic and geometric scene components. Therefore, we promote flow consistency at motion boundaries with an auxiliary second-order term implementing *focal* gradient matching:

$$\mathcal{L}_\nabla^x(u_2, u_2^*) = (1 - e^{-\nabla_x u_2 / \sigma}) \| \nabla_x u_2 - \nabla_x u_2^* \|_1, \tag{6}$$

where $u_2^* := \mathcal{H}_\theta(v_2)A^*$ and $\nabla_x$ is the spatial gradient along the $x$-axis of the image plane. Equivalently, we compute the gradient for the $y$-axis and the corresponding term $\mathcal{L}_\nabla^y$.

---

[2]If $A^*$ and $\mathcal{H}^*$ are the solutions, so are $cA^*$ and $\mathcal{H}^*/c$ for any $c \neq 0$.

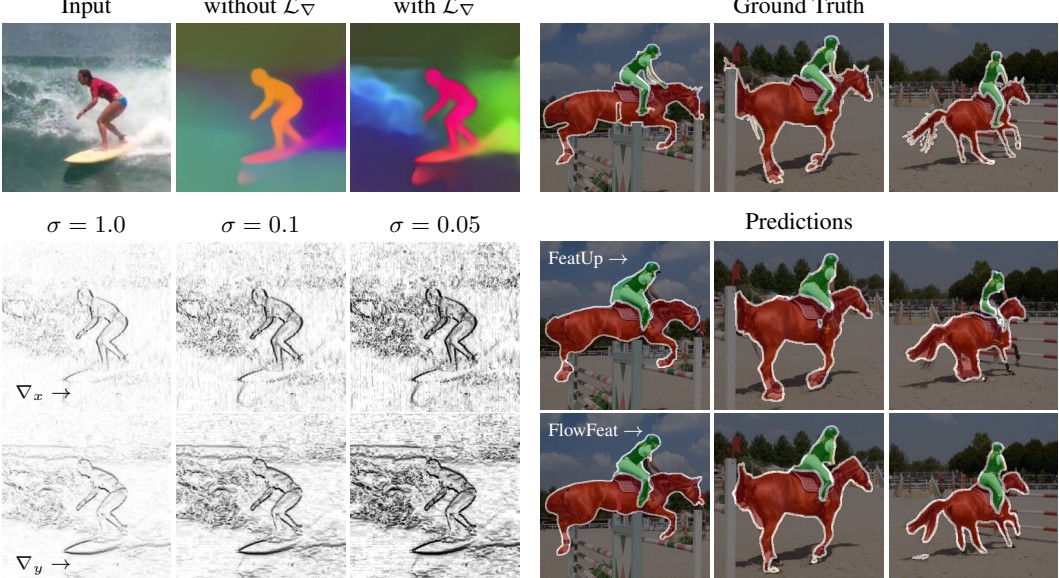

Figure 3: **Left: Focal gradient matching term $\mathcal{L}_\nabla$.** The first row visualises the first three PCA components of FlowFeat trained with and without the gradient term. Observe sharper feature boundaries with the use of the gradient term. Additionally, we found benefit in modulating the gradient difference with a hyperparameter $\sigma$, as defined in Eq. (6). The modulation with a lower $\sigma$ amplifies the effect of motion discontinuities (here, demonstrated for *image* gradients). **Right: Qualitative examples on VOS.** FlowFeat reveals finer details of the semantic masks compared to existing upsampling strategies, such as FeatUp [15].

Fig. 3 illustrates the effect of the gradient matching loss. As we also demonstrate empirically in Sec. 4.4, the gradient loss results in sharper feature maps (see the top row in Fig. 3). Note that the focal term in Eq. (6) enables modulation of the gradient loss at motion discontinuities. As the two bottom rows in Fig. 3 demonstrate, the hyperparameter $\sigma$ controls the degree of this modulation: a lower value of $\sigma$ results in sharper FlowFeat boundaries. However, a very low value of $\sigma$ may amplify the negative effect of inaccurate flow predictions, which can also exhibit flow discontinuities.

The total second-order flow reconstruction loss is simply a weighted sum:

$$\mathcal{L}_{\text{total}} = \mathcal{L}_\nabla + \lambda \mathcal{L}_{L1}, \tag{7}$$

where $\mathcal{L}_\nabla$ is the sum of $\mathcal{L}_\nabla^x$ and $\mathcal{L}_\nabla^y$, and $\lambda$ is a trade-off hyperparameter kept fixed across all models.

# 4 Experiments

We probe FlowFeat on three diverse tasks: video object segmentation (VOS), semantic segmentation and monocular depth prediction. Our goal is to demonstrate that FlowFeat offers substantial and consistent benefits across these downstream tasks as well as across backbone models, regardless of their pre-training strategy. Overall, we train FlowFeat on top of five backbone models: Masked Autoencoder (MAE) [18] based on ViT-B16, DINO [9] based on ViT-B16 and ViT-S16, and DINO2 [37] based on ViT-B14 and ViT-S14. As the decoder architecture and the only trainable component in FlowFeat, we use the DPT model [41], which is runtime-efficient (*cf*. Tab. 7, supp. material). The flow distillation relies on SEA-RAFT [52] based on ResNet-34. However, our ablation experiments in Sec. 4.4 with the older RAFT model [47] and unsupervised flow [44] show that this choice of the flow estimator is not critical. Furthermore, Fig. 6 illustrates the resilience of the training to inaccurate flow targets. We report the results for two FlowFeat variants. FlowFeat-*YT* trains on 3471 video sequences from YouTube-VOS (CC BY 4.0, [54]). For larger backbones, we train FlowFeat-*K* on Kinetics-400 dataset (CC BY 4.0, [23]) containing 147646 videos.[3] We compare our FlowFeat variants to the corresponding encoder model, as well as FeatUp [15], pre-trained on COCO-Stuff

---

[3]We exclude videos containing a montage of multiple clips to ensure temporal coherence.

| Method | Train Data | Linear Probing | | | Local KNN | | |
|---|---|---|---|---|---|---|---|
| | | $\mathcal{JF}$ | $\mathcal{J}_m$ | $\mathcal{F}_m$ | $\mathcal{JF}$ | $\mathcal{J}_m$ | $\mathcal{F}_m$ |
| V-JEPA [5] | VideoMix2M [5] | 49.0 | 46.1 | 51.9 | 56.7 | 55.6 | 57.8 |
| VideoMAE [48] | Kinetics | 43.3 | 40.9 | 45.8 | 55.1 | 54.6 | 55.6 |
| MAE-B16 [18] | ImageNet | 40.8 | 38.5 | 43.1 | 44.3 | 42.8 | 45.8 |
| +FlowFeat-K | +Kinetics | **53.8** | **50.1** | **57.5** | **59.1** | **57.3** | **60.8** |
| DINO-B16 [9] | ImageNet | 52.3 | 49.1 | 55.4 | 62.3 | 60.7 | 64.0 |
| +FlowFeat-YT | +YT-VOS | 55.5 | 52.5 | 58.5 | 64.0 | 62.7 | 65.3 |
| +FlowFeat-K | +Kinetics | **56.9** | **53.7** | **60.1** | **66.0** | **64.5** | **67.5** |
| DINO-S16 [9] | ImageNet | 49.6 | 46.8 | 52.4 | 61.5 | 59.9 | 63.1 |
| +FeatUp [15] | COCO-S | 52.4 | 49.6 | 55.2 | 63.7 | 62.4 | 64.9 |
| +FlowFeat-YT | +YT-VOS | 54.1 | 51.1 | 57.0 | 63.7 | 62.0 | 65.5 |
| +FlowFeat-K | +Kinetics | **56.2** | **52.9** | **59.5** | **66.5** | **64.5** | **68.4** |
| DINO2-B14 [37] | LVD* | 61.6 | 58.5 | 64.7 | 66.4 | 64.4 | 68.5 |
| +FlowFeat-YT | +YT-VOS | 65.7 | 62.2 | 69.2 | 69.0 | 66.9 | 71.2 |
| +FlowFeat-K | +Kinetics | **66.1** | **62.3** | **69.9** | **69.9** | **67.7** | **72.1** |
| DINO2-S14 [37] | LVD* | 57.5 | 54.2 | 60.7 | 65.1 | 63.7 | 66.6 |
| +FeatUp [15] | COCO-S | 60.5 | 57.4 | 63.6 | 65.5 | 65.0 | 66.1 |
| +LoftUp [20] | +SA1B [25] | 63.0 | 59.6 | 66.4 | 66.0 | 64.7 | 67.4 |
| +FlowFeat-YT | +YT-VOS | **65.8** | **62.0** | **69.7** | 67.6 | 65.6 | 69.6 |
| +FlowFeat-K | +Kinetics | 64.6 | 61.0 | 68.2 | **68.5** | **66.1** | **70.9** |

Table 1: **Video object segmentation (VOS) with linear probing and label propagation (local KNN) on DAVIS-2017 (val).** FlowFeat significantly improves the VOS accuracy of the baselines across all tested scenarios. It further outperforms previous and concurrent upsampling techniques (FeatUp [15] and LoftUp [20]). Pre-training FlowFeat on the larger Kinetics datasets tends to produce a stronger representation. LVD* refers to the distillation from a model pre-trained on LVD [37]. LoftUp [20] uses SAM, trained with mask supervision on SA1B [25].

(CC BY 4.0 / Flickr, [8]). Recall that FeatUp stacks multiple bilateral upsamplers and preserves the feature dimensionality. For instance, FeatUp yields representations with dimensionality 384 for ViT-S, whereas FlowFeat is more compact and has a fixed dimensionality of 128 across all variants. This allows us to evaluate FlowFeat in a complementary fashion to the backbone encoding by jointly fitting a high-resolution probe on FlowFeat and a low-resolution probe on the fixed encoder.

**Implementation details (see also Appendix B).** Training FlowFeat is computationally inexpensive. To train one model, we use a *single* GPU with 46GB of memory. The training proceeds with mini-batches of 128 images, input resolution $224 \times 224$ and AdamW optimiser [24, 31] with learning rate $10^{-4}$ and no weight decay. For the hyperparameters, we empirically set $\lambda = 0.1$, $\sigma = 0.1$ and $\gamma = 1.0$ and did not observe sensitivity to moderate deviations from these values. We train FlowFeat for 500 epochs on YouTube-VOS and for 100 epochs on Kinetics. In wall-clock time with one A40 GPU, the training takes only 24 hours and 3 days for YouTube-VOS and Kinetics, respectively.

## 4.1 Video object segmentation

We evaluate FlowFeat on semi-supervised video object segmentation (VOS) using 30 validation sequences from DAVIS-2017 (CC BY-SA 4.0, [40]). The task is to propagate the ground-truth annotation defined in the first frame to the rest of the video. Therefore, performing well on this task would indicate the capacity for temporal invariance as well as pixel-level semantic discrimination.

Previous evaluation protocols for VOS employ a variant of a localised k-nearest neighbour classifier [2, 21, 28], referred to as *local KNN* in the following. This probing technique is known to be brittle, exhibiting high volatility w.r.t. its hyperparameters [34]. For consistency with previous work, we stick to the implementation of local KNN provided by Caron et al. [9]. However, we additionally evaluate VOS with *linear probing*, as the more established and interpretable technique in representation learning [11, 16]. Linear probing extends seamlessly to the VOS task. Specifically, for each video, we train a linear classifier using the ground-truth segmentation provided for the first frame. We apply the linear classifier to the remaining frames to obtain the segmentation result. For both probing strategies – linear probing and local KNN – we compute the mean region similarity $\mathcal{J}_m$, the mean contour-based accuracy $\mathcal{F}_m$ and their mean $\mathcal{JF}$. Tab. 1 reports the results. Across all pre-training methods and metrics, FlowFeat achieves a consistent and substantial improvement in VOS accuracy. The benefit is especially significant for MAE-B16, where FlowFeat improves the baseline by staggering $13.0\%$ / $14.8\%$ $\mathcal{JF}$ with linear probing / local KNN. However, FlowFeat also surpasses stronger baselines, *e.g.* DINO2-B14 ($+4.5\%$ / $+3.5\%$ $\mathcal{JF}$) and FeatUp ($+3.8\%$ / $+2.8\%$ $\mathcal{JF}$ for DINO-S16 and $+5.3\%$ / $+3.0\%$ for DINO2-S14 $\mathcal{JF}$). As illustrated in Fig. 3 (right), the improvement is especially pronounced at the object boundaries. FeatUp enhances VOS accuracy for both baselines (DINO-S16 and DINO2-S14), but these improvements are more modest. FeatUp also struggles with inputs of higher resolution, introducing static feature artefacts, see the supplemental videos and further analysis

Table 2: **Probing semantic segmentation and monocular depth.** On COCO-Stuff 2017 (val), FlowFeat advances the segmentation quality across all baselines as well. A lightweight refinement using FlowFeat++ (numbers in parentheses) further boosts the accuracy without any parameter training. On NYUv2 (val), FlowFeat significantly improves the depth accuracy across all pre-trained encoders – in contrast to FeatUp, which struggles to improve upon its baselines.

| Method | Semantic Segmentation | | | | Depth Estimation | | | |
|---|---|---|---|---|---|---|---|---|
| | mIoU ↑ | | pAcc ↑ | | RMSE ↓ | $\delta_1$ ↑ | $\delta_2$ ↑ | $\delta_3$ ↑ |
| MAE-B16 [18] | 46.0 | | 71.5 | | 0.4534 | 83.68 | 96.98 | 99.28 |
| + FlowFeat-K | **47.2** | | **72.9** | | **0.4400** | **84.43** | **97.18** | **99.35** |
| DINO-B16 [9] | 46.1 | | 72.0 | | 0.4287 | 86.15 | 97.61 | 99.47 |
| + FlowFeat-K | **48.2** | | **73.7** | | **0.4176** | **86.87** | **97.71** | **99.50** |
| FeatUp – DINO-S16 [15] (++) | 41.6 | (42.1) | 69.5 | (69.9) | 0.4624 | 83.54 | 96.90 | 99.32 |
| DINO-S16 [9] | 39.6 | | 67.5 | | 0.4634 | 83.60 | 96.94 | 99.32 |
| + FlowFeat-YT (++) | **44.7** | **(45.9)** | **71.4** | **(72.5)** | **0.4410** | **85.26** | 97.17 | 99.30 |
| + FlowFeat-K (++) | 44.2 | (45.4) | 71.3 | (72.3) | 0.4422 | 84.81 | **97.19** | **99.37** |
| DINO2-B14 [37] | 58.1 | | 78.0 | | 0.3091 | 94.14 | 99.32 | 99.89 |
| + FlowFeat-K | **60.4** | | **79.8** | | **0.2791** | **95.55** | **99.52** | **99.93** |
| FeatUp – DINO2-S14 [15] (++) | **58.3** | (58.5) | **79.1** | (79.2) | 0.3207 | 93.29 | 99.18 | 99.86 |
| DINO2-S14 [37] | 56.2 | | 77.3 | | 0.3294 | 92.97 | 99.11 | 99.85 |
| + FlowFeat-YT (++) | 58.0 | (59.4) | 78.7 | (79.7) | 0.3072 | 93.91 | 99.25 | 99.86 |
| + FlowFeat-K (++) | 58.1 | **(59.6)** | 78.9 | **(79.9)** | **0.3061** | **94.12** | **99.31** | **99.88** |

in Tab. 6. Similarly, the contemporaneous work, LoftUp [20], achieved inferior accuracy despite the implicit leverage of vast mask supervision via SAM [25]. Video-based models, such as V-JEPA [5] and VideoMAE [48], are also remarkably ineffective.

Overall, the improvements on VOS metrics provide compelling evidence that FlowFeat encapsulates a high degree of temporal invariance and feature detail, with complementary properties to the encoder representation. Furthermore, the larger Kinetics dataset tends to produce a stronger variant of FlowFeat. This observation indicates that FlowFeat has the promising capacity to scale with the ever-increasing volume of real-world videos.

## 4.2 Semantic segmentation

We follow the setting of FeatUp [15] and use COCO-Stuff 2017 with $C = 27$ coarsely annotated categories [8]. Since FlowFeat focuses on motion patterns rather than global semantic alignment, it may lack consistent semantic structure across images; therefore, we employ attention probing [5] to derive image-specific class prototypes. In more detail, we define $C = 27$ learnable queries attending the FlowFeat representation with a single layer of cross-attention. We freeze the models and train the probes on $256 \times 256$ centre crops using the cross-entropy loss. Additionally, we demonstrate that FlowFeat can further boost the segmentation accuracy with a simple adaptation of a lightweight post-processing technique. Concretely, we adapt the local mask refinement strategy (PAMR) [1], but leverage FlowFeat instead of the image intensity to refine the segmentation result. Note that such a refinement is not possible by the use of the probes alone due to their feed-forward nature. We refer to this straightforward extension as FlowFeat++. Appendix B.2 provides further details.

Tab. 2 reports the mean pixel accuracy and the mean IoU. The results align with our observations in VOS experiments: FlowFeat boosts the accuracy across all baseline models. Particularly notable are the improvements w.r.t. smaller models. For example, FlowFeat surpasses DINO-S16 by $5.1\%$ and $4.6\%$ with FlowFeat-YT and FlowFeat-K, respectively. Without the refinement, FlowFeat performs competitively with FeatUp [15] based on DINO2-S14 and outperforms it for DINO-S16. Furthermore, the FlowFeat-based refinement significantly enhances the segmentation quality. For example, FlowFeat-K++ improves over FlowFeat-K by a notable margin of $1.5\%$ mIoU. By contrast, FeatUp does not profit from the refinement as much.

Fig. 5 visualises the segmentation results for the DINO2-S14 backbone, with and without the refinement. Initial predictions of the probes are coarse and lack detail, especially around object boundaries. Leveraging the high-resolution FlowFeat representation (visualised with PCA), the refinement leads to sharper mask alignment with image boundaries.

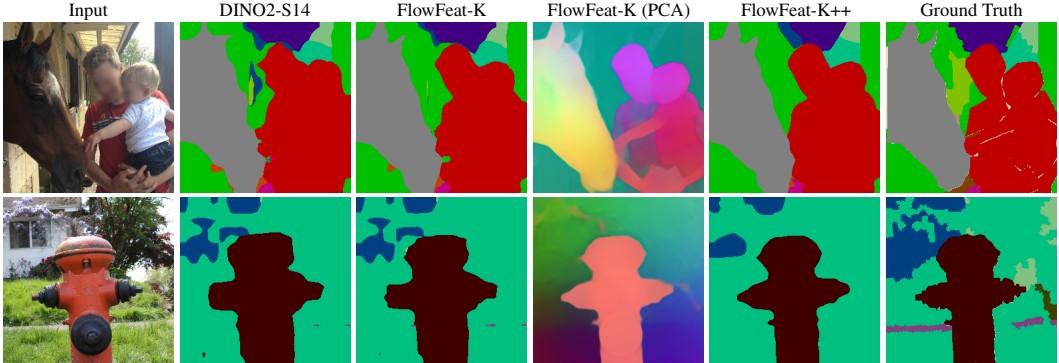

| Input | DINO2-S14 | FlowFeat-K | FlowFeat-K (PCA) | FlowFeat-K++ | Ground Truth |

Figure 5: **Semantic segmentation and post-hoc refinement (++) with FlowFeat.** The segmentation masks from FlowFeat exhibit a high level of boundary accuracy. The FlowFeat representation, visualised with PCA, identifies prominent scene elements with a fine-grained detail. A lightweight post-hoc refinement (FlowFeat-K++), based on PAMR [1], leverages the pairwise pixel similarity embedded by FlowFeat (instead of image intensities) to improve the results further.

In summary, FlowFeat provides a significant boost also for downstream semantic tasks. The feature representation offers a high degree of spatial detail and also lends itself well to lightweight post-processing without the need for additional training.

## 4.3  Monocular depth estimation

We evaluate FlowFeat on a geometric task, monocular depth estimation, using NYUv2 [36], following the evaluation protocol of Banani et al. [4]. Similar to the VOS setting, we compare FlowFeat against self-supervised backbones: MAE [18], DINO [9], and DINO2 [37]. As in semantic segmentation, we use attention probing [5] to extract the depth-specific prototypes from FlowFeat. Specifically, we utilise the AdaBins [6] formulation that quantises the depth into 256 bins. The depth value is a weighted sum of the predicted distribution across the bins and the corresponding depth value of the bin. Following Banani et al. [4], we optimise the model using a weighted combination of the scale-invariant depth loss [6] and a gradient loss. Appendix B provides further details on probe implementation and training.

Adhering to the setting of Banani et al. [4], we train the probes on the NYUv2's training set (24231 images) and evaluate the models on $480 \times 480$ centre crops of the 1449 validation images [36]. As the standard depth metrics, we report the root-mean-squared error (RMSE) and the inlier rates at three thresholds. Specifically, $\delta_i$ corresponds to the fraction of depth predictions $d$ satisfying $\max(d/d^*, d^*/d) < 1.25^i$ w.r.t. the ground-truth $d^*$.

Tab. 2 summarises the quantitative results. In line with our observations on VOS and semantic segmentation, FlowFeat achieves a notable boost across all baseline models. For example, FlowFeat-K reduces RMSE w.r.t. the DINO-S16 model by $0.051$ and increases the $\delta_1$ by $3.16\%$. By contrast, we did not observe benefit from the high-resolution FeatUp, which appears to be biased towards the pre-training resolution of $224 \times 224$. Notably, we did not observe such a detrimental bias in FlowFeat.

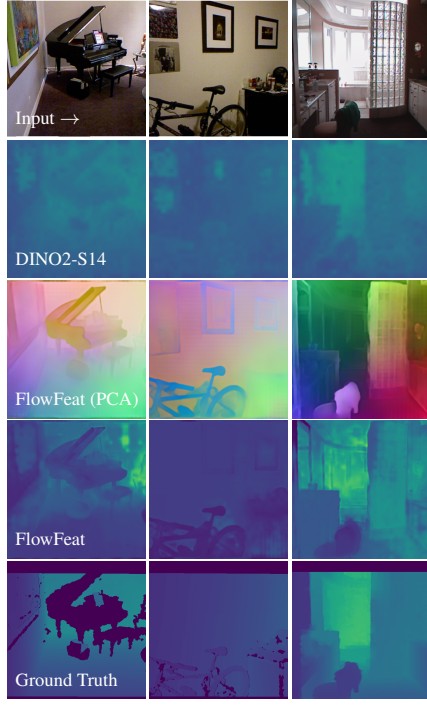

Figure 4: **Depth probing.** FlowFeat significantly improves depth estimates for challenging elements, such as non-Lambertian surfaces (*e.g.* left, the piano), intricate structures (*e.g.* middle, the bicycle), and under- and oversaturated image areas (*e.g.* right, a bathroom).

Table 3: **Ablation study on DAVIS-2017 (val).** Following Sec. 4.1, we perform linear probing on VOS in a set of ablation experiments. The $\Delta$ reports the absolute difference in the corresponding metric w.r.t. the baseline. The ridge regularisation in FlowFeat is crucial, but the choice of the flow estimator is not instrumental.

| Baseline: DINO2-S14 | $\mathcal{JF}$ | $/\Delta$ | $\mathcal{J}_m$ | $/\Delta$ | $\mathcal{F}_m$ | $/\Delta$ |
|---|---|---|---|---|---|---|
| +Random DPT | 58.7 | | 55.2 | | 62.2 | |
| +FlowFeat-YT | **65.8** | | **62.0** | | **69.7** | |
| (a) naïve | 56.7 | −9.1 | 52.8 | −9.2 | 60.5 | −9.2 |
| (b) $\gamma = 0.001$, cf. Eq. (4) | 58.2 | −7.6 | 54.9 | −7.1 | 61.5 | −8.2 |
| (c) w/o $\mathcal{L}_\nabla$, cf. Eq. (6) | 64.3 | −1.5 | 61.0 | −1.0 | 67.7 | −2.0 |
| (d) $\mathcal{L}_{L2}$ | 63.3 | −2.4 | 59.8 | −2.2 | 67.0 | −2.7 |
| (e) w/o $\mathcal{L}_{L1}$, cf. Eq. (5) | 65.3 | −0.5 | 61.6 | −0.4 | 69.0 | −0.7 |
| (f) RAFT | 65.2 | −0.6 | 61.6 | −0.4 | 68.8 | −0.9 |
| (g) SMURF (unsupervised) | 64.1 | −1.7 | 60.7 | −1.3 | 67.5 | −2.2 |
| (h) temp. window $\times$ 2 | 65.5 | −0.3 | 62.1 | +0.1 | 68.9 | −0.8 |
| (i) next frame only | 65.8 | 0.0 | 62.2 | +0.2 | 69.4 | −0.3 |

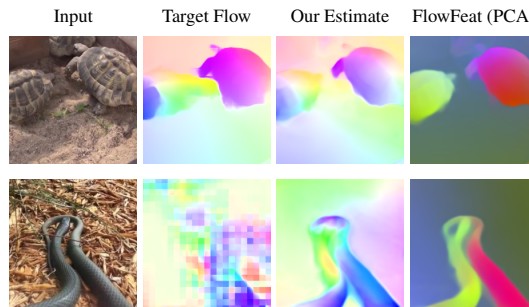

Input    Target Flow    Our Estimate    FlowFeat (PCA)

Figure 6: **Resilience to inaccurate flow targets.** Despite inaccurate and even artefact-prone predictions from optical flow networks, FlowFeat learns a reasonable flow approximation without compromising the feature representation.

We visually inspect the results in Fig. 4 for the DINO2-S14 backbone. In contrast to the low-quality depth estimates extracted from the frozen encoder, FlowFeat representation exhibits an impressive degree of fine-grained detail. This is indeed noteworthy, considering that FlowFeat originates from video data and was not trained for such static scenes. FlowFeat is also robust to under- and oversaturated image areas (cf. Fig. 4, the rightmost column) and infers highly plausible depth where the ground truth is not available due to surface specularity (cf. Fig. 4, the piano).

In summary, the results suggest that the motion profiles embedded by FlowFeat provide strong geometric cues. FlowFeat not only enhances the depth awareness across all baselines, but also scales compellingly with the increased amount of video data for pre-training: FlowFeat-K outperforms the less data-intensive FlowFeat-YT across virtually all settings and metrics.

## 4.4  Ablation study

We conduct an ablation study of FlowFeat on the DAVIS-2017 (val) benchmark. The study follows the evaluation protocol with linear probing from Sec. 4.1. Using ViT-S14 backbone pre-trained with DINO2 [37], we train a number of FlowFeat configurations on YouTube-VOS [54]. Tab. 3 reports the results. As a sanity check, we verify that a randomly initialised DPT decoder has virtually no effect on the VOS accuracy (cf. Tab. 3 in grey). Similarly, (a) naïvely fitting optical flow with a *single* linear layer (instead of a distribution) is futile. Next, (b) we verify the benefit of estimating the optimal operator $A^*$ in Eq. (5) with ridge regression. To compare with the baseline setting of $\gamma = 1$ (cf. Eq. (4)), we trained the model with $\gamma = 10^{-3}$. We found the training numerically unstable with $\gamma = 0$. Thus, setting $\gamma$ to $10^{-3}$ provides a reasonable approximation to removing the effect of L2-regularisation on the linear operator $A$. In this case, the VOS accuracy drastically deteriorates across all metrics, which justifies the crucial need for ridge regularization. (c) We train FlowFeat without the second-order term, $\mathcal{L}_\nabla$ in Eq. (6). A drop in the downstream accuracy (e.g. $-2.0\%$ $\mathcal{F}_m$) suggests that FlowFeat exploits motion boundaries in its representation, in line with the established view that motion boundaries are strong semantic cues. (d) Replacing the $L_1$ reconstruction loss by $L_2$ distance in Eq. (5) reduces $\mathcal{J}\&\mathcal{F}$ from $65.8\%$ to $63.4\%$, supporting the robustness of $L_1$ in comparison to $L_2$. Next, (e) we explore a configuration without the L1 reconstruction term by setting $\lambda := 0$ in Eq. (7). Surprisingly, the drop in accuracy is not substantial. This suggests that the training process can succeed with the gradient loss alone. However, we observed that including the L1 reconstruction term tends to improve the convergence speed consistently across all models. In the next experiments (f,g), we replace SEA-RAFT [52] with the RAFT model [47] and the unsupervised SMURF [44], respectively. The drop in the VOS accuracy is not significant, which demonstrates that obtaining FlowFeat is not sensitive to a specific choice of the flow model. Fig. 6 further examines the training resilience to inaccurate target flow. In both examples, the optical flow from the pre-trained network is inaccurate (even artefact-prone), yet it has no visible effect on the quality of FlowFeat. Curiously, the second example in Fig. 6 also reveals one limitation of FlowFeat: the apparent motion of the tail and the head of the snake have opposite directions, which decouples their feature representation.

Finally, *(h,i)* we test two strategies for sampling frame pairs from a video in Tab. 3. Our base configuration uses a temporal window of 5 frames and can select frame $t'$ either from the past or the future. We increase this temporal window to 9 *(h)*, which leads to a slight deterioration of VOS accuracy – presumably due to the more challenging estimation of optical flow. The setting *(i)* selects the immediate next future frame as $t'$. Here, the VOS accuracy barely changes. This implies that *(i)* FlowFeat does not simply overfit apparent motion (compare to *(a)*), and *(ii)* the motion samples *across the dataset*, not just a temporal window, play a more critical role in embedding motion profiles.

**Further study.** Our further analysis, provided in the supplemental material, shows that: *(1)* FlowFeat scales well with the input resolution, further improving the VOS accuracy when the input resolution is doubled (*cf*. Tab. 6); *(2)* the accuracy gains from FlowFeat do not arise merely from higher resolution of the feature map per se, but from its complementary motion-derived properties (*cf*. Tab. 5a); *(3)* FlowFeat also scales effectively to larger transformer models (*e.g*. ViT-L) (*cf*. Tab. 5b).

## 5   Limitations

**Application scope.** FlowFeat relies on a pre-trained optical flow network and video data for training. It assumes either brightness constancy in the video stream or availability of synthetic data for pre-training the optical flow model. While these assumptions generally hold for standard RGB videos, they may not apply in other domains, such as medical imaging (*e.g*. MRI, CT), thermal imaging or low-light scenarios.

**Frozen backbone.** Recall that training FlowFeat involves updating only the decoder parameters, while keeping the encoder parameters fixed. Consequently, the encoder representation imposes an upper bound on FlowFeat's downstream accuracy, especially in terms of high-frequency content. Although we have shown that FlowFeat generalises across widely used self-supervised encoders, such as MAE [18], DINO [9], DINOv2 [37] and across different model capacities, FlowFeat may be less effective with backbones that underrepresent high-frequency details in their intermediate feature maps.

| Model | Person | Wall | Landscape | Vegetation | Ground |
|---|---|---|---|---|---|
| DINO-S16 | 69.3 | 46.5 | 43.9 | 65.5 | 33.3 |
| + FlowFeat -YT | **75.6** | **50.0** | **50.8** | **69.9** | **37.0** |
| DINO-B16 | 72.9 | 51.4 | 51.0 | 70.3 | 38.9 |
| + FlowFeat -K | **77.8** | **52.9** | **53.1** | **71.7** | **39.6** |
| DINOv2-S14 | 76.9 | 57.6 | 59.2 | 71.0 | 44.6 |
| + FlowFeat -YT | **81.7** | **59.3** | **60.5** | **73.0** | **45.7** |
| DINOv2-B14 | 77.0 | 59.2 | 59.6 | 70.3 | 44.9 |
| + FlowFeat -K | **83.0** | **61.7** | **61.3** | **72.3** | **45.1** |
| MAE-B16 | 72.2 | 50.8 | 52.7 | 66.9 | 36.1 |
| + FlowFeat -K | **78.6** | **51.3** | **53.7** | **69.9** | **38.8** |

Table 4: **Semantic segmentation accuracy on COCO-Stuff (IoU, %).** As expected from motion parallax, the gains on (potentially) dynamic classes (*e.g*. "person") are larger compared to that of typical background categories (*e.g*. "vegetation"). Nevertheless, FlowFeat leads to a consistent segmentation improvement across *all* categories.

**Motion bias.** Owing to its training approach, FlowFeat tends to emphasise image regions with larger magnitudes of expected motion, typically corresponding to foreground dynamic objects, relative to the static background areas. To quantitatively assess this behaviour, we report per-category IoU scores on COCO-Stuff in Tab. 4, following the probing protocol described in Sec. 4. We observe that the improvement on the "person" category is indeed more pronounced than for static classes. Nevertheless, FlowFeat yields consistent accuracy gains across all categories, regardless of whether they are static or dynamic in nature.

## 6   Conclusion

We presented FlowFeat, a pixel-dense and versatile representation embedding motion profiles. Our experiments provide compelling evidence that FlowFeat enhances the representation power of pre-trained encoders across all downstream tasks considered in our study. Specifically, FlowFeat possesses temporal consistency and exhibits a remarkable level of spatial detail, encompassing semantic and geometric cues without explicit supervision. More broadly, our work addresses motion stochasticity in a principled fashion, revealing a powerful synergy between optical flow networks and large video datasets. FlowFeat takes a significant step towards label-efficient and versatile models for high-precision tasks, such as image-based 3D reconstruction, object-level segmentation and tracking.

**Acknowledgements.** This work was supported by the ERC Advanced Grant SIMULACRON and DFG project CR 250/26-1 "4D-YouTube". NA thanks Junhwa Hur and Jochen Gast for their valuable feedback.

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
