# OpenReview forum: "FlowFeat: Pixel-Dense Embedding of Motion Profiles"
_NeurIPS.cc/2025/Conference — NeurIPS 2025 spotlight_

### Official Review · Reviewer_zphZ · 2025-06-27

**Clarity:** 2
**Significance:** 2
**Originality:** 2
**Rating:** 4
**Confidence:** 4

**Summary:**

The paper introduces a new feature representation named FlowFeat, a multi-task feature representation learned through a novel distillation technique that embeds distributions of plausible apparent motion (motion profiles). FlowFeat leverages optical flow networks and diverse video data in a self-supervised framework to produce pixel-level features with high spatial precision and temporal consistency. The authors evaluate FlowFeat on three dense prediction tasks—video object segmentation (VOS), monocular depth estimation, and semantic segmentation—demonstrating consistent improvements over state-of-the-art encoders and upsampling strategies like FeatUp. Key contributions include:
1.A self-supervised training framework combining optical flow networks and video data to embed motion profiles.
2.Empirical validation showing FlowFeat enhances representation quality across multiple tasks and backbone architectures.

**Questions:**

Please see the weaknesses

**Ethical Concerns:**

["NO or VERY MINOR ethics concerns only"]

**Final Justification:**

Thanks for the rebuttal. Some of my concerns are addressed. Therefore, I would like to raise my score. Please include intuition analysis and discussions of limitations in the revision. Thanks

**Limitations:**

Limitations are discussed, but more are needed to further validate the performances of FlowFeat with respect to different aspects, e.g. scene parallax, occlusions, textureless areas etc.

**Quality:**

2

**Strengths And Weaknesses:**

Pros.
The idea of distilling motion profiles via optical flow networks is innovative and addresses a clear gap in dense feature learning.  FlowFeat is computationally efficient, trainable on a single GPU, and scales well with input resolution without fine-tuning. The framework is shown to be resilient to inaccurate flow estimates and works with unsupervised flow networks.

Cons.
Three tasks are selected for evaluation, e.g., VOS, MDE, and semantic segmentation. However, one of the most important task, optical flow estimation, are omitted. Given that the flowfeat is computed from video frames, optical flow is the natural task that should be evaluated.

While the empirical results are strong, the paper lacks a deeper theoretical analysis of why motion profiles lead to better feature representations. For instance, how does the distribution of linear transformations generalize across tasks?

The comparison to FeatUp is limited; why does FlowFeat outperform it? Is it purely due to motion cues, or are there architectural differences?

The discussion of limitations (e.g., non-rigid motion handling in Fig. 6) is brief. How does FlowFeat perform on highly dynamic or occluded scenes? Are there failure modes not shown?

Can you provide more intuition or analysis for why motion profiles lead to better features? For example, is there a connection to equivariance or invariance in representation learning?

---

> ### Author Rebuttal · Authors · 2025-07-29
>
> Thank you for your time and your valuable feedback.
> Please note that NeurIPS will provide an additional page for the camera-ready version. We plan to use the allocated space to expand our analysis of limitations and provide more intuition behind the approach. Thank you for these suggestions.
>
> We clarify and respond to your comments below.
>
> > **[...] optical flow is the natural task that should be evaluated.**
>
> Observe that FlowFeat is a *monocular* model – much like DINO and MAE.
> Since FlowFeat learns motion patterns from a pre-trained optical flow network, we do not expect -- and do not claim -- that FlowFeat will outperform the teacher network on flow estimation.
>
> Instead, we focus on cross-task generalisation, as the common practice in representation learning. Therefore, we use benchmarks that are very different from optical flow estimation.
> Specifically, we evaluate FlowFeat on **three benchmarks**, train **9 model variants** of FlowFeat and test **2** datasets for pre-training (YouTube-VOS and Kinetics). We believe that this evaluation scope and the consistency of the results across all experiments provides sufficient empirical evidence for our technical contribution.
>
> > **[...] the paper lacks a deeper theoretical analysis of why motion profiles lead to better feature representations. For instance, how does the distribution of linear transformations generalize across tasks?**
>
> A theoretical analysis of our work is currently infeasible due to the fact that we are training a deep model on weakly curated video data with stochastic optimisation. However, the intuition behind generalisation of FlowFeat is the following.
>
> The least-squares solution A* will be specific to the image pair, on which it was computed, and it will not generalise to fit the optical flow of another frame pair.
> However, by training the model on many frame pairs -- implying many different linear projections A* -- our embeddings learn to satisfy the induced distribution of those linear transformations w.r.t. the apparent motion. As a result, the network must encode non-trivial feature maps embedding plausible apparent motion.
>
> > **[...] why does FlowFeat outperform [FeatUp]? Is it purely due to motion cues, or are there architectural differences?**
>
> This is a valid concern. However, we compare to FeatUp while ensuring the same backbone architecture. Therefore, the observed improvements come directly from the motion cues.
>
> > **[...] How does FlowFeat perform on highly dynamic or occluded scenes? Are there failure modes not shown?**
>
> The VOS benchmark contains some highly dynamic and occluded scenes, and is broadly adopted in our community for these reasons. As shown quantitatively (c.f. Tab. 1), FlowFeat consistently improves the baseline accuracy.
>
> As shown in some qualitative results, FlowFeat allocates less capacity to background elements. Quantitatively, the improvement on background areas is more modest in comparison to the improvement on the foreground regions. Nevertheless, FlowFeat demonstrates consistent benefits in terms of segmentation and depth metrics.
>
>
> > **Can you provide more intuition or analysis for why motion profiles lead to better features? For example, is there a connection to equivariance or invariance in representation learning?**
>
> Intuitively, the field of apparent motion is highly structured -- recall the Gestalt principle “what moves together belongs together”. Now, let us consider constant motion as an example. In this case, approximating constant flow (no motion boundaries) is trivial both in terms of A* and the feature map. However, to approximate complex motion patterns with a linear projection A*, the network must learn a non-trivial feature map, which groups pixels that tend to have similar motion. As we train the network on a distribution of A*’s (induced by randomly sampling frame pairs), the network will learn a highly structured and spatially meaningful feature representation.
>
> Indeed, FlowFeat is based on invariance of A* for overlapping image crops: the observed motion in one image crop must be sufficient to explain the motion in its adjacent crop.
>
>
> **Thank you for reading our response. As we did not recognise any critical concerns in the initial review, we hope for a more supportive recommendation.**

---

> > ### Comment · Reviewer_zphZ · 2025-08-08
> >
> > Thanks for the rebuttal. Some of my concerns are addressed. Therefore, I would like to raise my score. Please include intuition analysis and discussions of limitations in the revision. Thanks

---

> > > ### Author Response · Authors · 2025-08-08
> > >
> > > Thank you again for your feedback and for reading our response.

---

### Official Review · Reviewer_2G2T · 2025-06-27

**Clarity:** 3
**Significance:** 3
**Originality:** 3
**Rating:** 5
**Confidence:** 3

**Summary:**

The paper proposes FlowFeat, a method that, based on an off-the-shelf generic visual representation (which are typically low-resolution), predicts high-resolution generic features using a decoder. Supervision for the decoder is primarily obtained via optical flow between frames from open-set videos, resulting in a high-resolution, dense supervision signal. The learned representation, starting from multiple different backbones, is evaluated across multiple different downstream tasks and demonstrates broad improvements over the respective base representation's performance.

**Questions:**

1. Regarding W1: Is this the case? If so, why not pass the image to the model in addition to the features? If not, this should be made clear throughout the paper (Fig. 2, Sec. 3, ...).
2. What happens if the representations obtained from FlowFeat are downsampled to the same resolution as the base model's representations? Evaluating in this setting as well would show whether the quantitative gains stem from increased resolution or from improved data representation (i.e., making the data that was already present in the base embedding more accessible for linear probing on the given tasks).
3. Regarding W2: Could the authors please elaborate on the motivation behind these choices, and provide ablations compared to intuitive baselines (e.g., L2 loss, no weighting, ...)?
4. Are the representations directly related to the base encoder's representations in any way?

Addressing these questions (and, ideally, also the other weaknesses mentioned) would help clarify what the method actually achieves and demonstrate its significance. I'm willing to raise the score to accept if the authors convincingly address these points and other reviewers do not uncover major unaddressed weaknesses.

**Ethical Concerns:**

["NO or VERY MINOR ethics concerns only"]

**Final Justification:**

As detailed in the strengths section, I consider the work interesting and valuable to the community. My initial rating of 4 stemmed from some open questions & weaknesses, which have mostly been resolved during the rebuttal, hence my raised score.

Overview of weaknesses & questions raised in original review
I consider mostly/fully resolved (Y), partially resolved (~), and unresolved (X):

| W1   | W2   | W3   | W4   | W5   | Q1     | Q2   | Q3     | Q4   |
| ---- | ---- | ---- | ---- | ---- | ------ | ---- | ------ | ---- |
| ~    | Y    | Y    | Y    | Y    | Y [W1] | N    | Y [W2] | Y    |

Additional details:

- W1: primarily about presentation, not a major weakness
- Q2: authors promised suggested eval for later, did not (yet) deliver during rebuttal, so unclear what result will be

**Limitations:**

Limitations are touched upon only very briefly and spread throughout the paper rather than being consolidated in a single place. Consolidating them into a limitations paragraph in the conclusion and extending upon them would improve the paper.
The PCA visualizations also appear to place a significant emphasis on moving objects, thereby obscuring the visibility of background elements, unlike other approaches. It would be worth discussing whether this constitutes a limitation of the approach and specific downstream capabilities, or whether it might simply be an artifact of the PCA visualization.

**Quality:**

3

**Strengths And Weaknesses:**

**Strengths**

The proposed method is relatively simple yet appears to be effective. While I'm not sure that describing the representation as embedding "statistical motion patterns" is a very fitting description, the idea of predicting a representation that captures which parts of the scene could move together and then solving for the actual motion to obtain supervision is interesting.
The experiments cover different tasks (video object/semantic segmentation and monocular depth) and show strong performance across them, outperforming the base representation across multiple standard off-the-shelf backbones.

**Weaknesses**
1. As far as I understood the paper, the feature decoder does not receive a copy of the image. This would mean that its success in providing high-resolution features is wholly dependent on the source features containing high-resolution image information, unlike FeatUp, which accounts for the full-resolution image. This appears to be a significant potential weakness of the method to me.
2. The authors make some non-obvious design choices, such as using an L1 loss instead of an L2 loss in Eq. 5 and the formulation of the weighting term in Eq. 6, without explaining them in detail or providing ablation studies.
3. Evaluations are only performed on small S- and B-scale models. It would be very valuable to demonstrate that the presented findings also generalize to larger models commonly used for practical applications.
4. Comparisons with FeatUp in Tab. 2 lack evaluation with the refinement stage for FeatUp and some base models, which should be included for a fair comparison.
5. Sec. 2 and 3 lack any structuring into subsections and/or paragraphs with headings, making them unnecessarily hard to read. Other sections have similar problems, but less severe. Introducing some additional structure could make the paper significantly more accessible.

---

> ### Author Rebuttal · Authors · 2025-07-29
>
> Thank you for your thoughtful and encouraging feedback.
>
> > **[...] the feature decoder does not receive a copy of the image.**
>
> We agree that this could be a valid concern. However, the DPT architecture, which FlowFeat uses, has skip connections to intermediate layers of the encoder, from which high-frequency information can be restored (despite the constant resolution in transformers). We kindly refer to Ranftl et al., [41] for architecture details and analysis. As can be seen from our qualitative examples (c.f. Figs. 1,3,4,5), FlowFeat exhibits fine-grained boundaries, suggesting no difficulty in recovering high-frequency information.
>
> > **The authors make some non-obvious design choices, such as using an L1 loss instead of an L2 loss in Eq. 5 and the formulation of the weighting term in Eq. 6**
>
> We actually experiment with the two terms in our ablation study, please refer to Tab. 6, (c,d).
>
> Since our training signal is an (potentially inaccurate) optical flow estimate, L1 distance is a natural choice due to its robustness in comparison to the L2 loss.
> We have verified this intuition by replacing the L1 training loss with the L2 loss. The results below, obtained on the VOS benchmark with DINO2-S14, confirm that L1 loss is a more appropriate choice.
>
> | Loss | J&F-Mean | J-Mean | F-Mean |
> |-----------|-----------|-----------|-----------|
> | L1 (baseline) | **65.8** | **62.0** | **69.7** |
> | L2 | 63.4 | 59.8 | 67.0 |
>
> > **[...]  It would be very valuable to demonstrate that the presented findings also generalize to larger models commonly used for practical applications.**
>
> Please note that demonstrating consistent benefits across 5 commonly used backbone architectures already provides strong empirical support for our claims. Nevertheless, we have now experimented with larger models, DINOv2 ViT-L and MAE ViT-L, and report the results on VOS with linear probing. As expected, FlowFeat (pre-trained on YouTube-VOS) consistently and significantly improves the baselines, and surpasses the VOS accuracy of smaller models. These results suggest that FlowFeat also generalises to larger models.
>
> | Model | J&F-Mean | J-Mean | F-Mean |
> |-----------|-----------|-----------|-----------|
> | DINOv2 ViT-L | 59.4 | 55.8 | 63.0 |
> | +FlowFeat-YT | **66.9** | **63.4** | **79.4** |
>
> | Model | J&F-Mean | J-Mean | F-Mean |
> |-----------|-----------|-----------|-----------|
> | MAE ViT-L | 46.7 | 44.4 | 49.0 |
> | +FlowFeat-YT | **55.4** | **52.0** | **58.9** |
>
>
> > **Comparisons with FeatUp in Tab. 2 lack evaluation with the refinement stage for FeatUp and some base models, which should be included for a fair comparison.**
>
> Please note that FeatUp employs a stack of bilateral solvers; PAMR, at best, only approximates the bilateral solver with multiple local operations. Moreover, as can be seen in Fig. 1, FeatUp tends to produce feature maps with inaccurate or “smudgy” boundaries. The refinement with PAMR will correspondingly "smudge" the segmentation masks, if used with the FeatUp’s feature maps, degrading the segmentation accuracy.
>
> As a remark, this is related to the earlier concern on high-frequency information. If FlowFeat lacked the high-frequency detail, the refinement would have no benefit. The results in Tab. 2 demonstrate the opposite.
>
> > **Sec. 2 and 3 lack any structuring into subsections and/or paragraphs with headings [...]**
>
> Thank you for the great suggestion. We will improve readability of Sec. 2 and 3 by structuring the text into run-in headings.
>
> > **Q.1 Regarding W1: Is this the case? [...]**
>
> As we discussed above – not exactly. The DPT decoder has skip connections, which allows FlowFeat to recover high-frequency detail, confirmed experimentally and qualitatively.
>
> > **Q.2 What happens if the representations obtained from FlowFeat are downsampled to the same resolution as the base model's representations? [...]**
>
> This is an interesting idea.
> However, our experimental setting already accounts for the possibility of ‘mere’ upsampling of the feature representation in two ways.
>
> First, we compare to FeatUp, which performs upscaling with a bilateral solver without computing a new representation. Intuitively, the FeatUp features remain in the ‘convex hull’ of the original representation available in the backbone. FlowFeat provides improved downstream accuracy on virtually all tasks (without refinement), which shows that the gains do not stem from the increased resolution alone.
>
> Second, in the local KNN setting of VOS evaluation (see Tab. 1), we keep the same hyperparameters of label propagation (e.g. the size of the local window) across all models (c.f. ll. 50-54, supp. material). This requires us to *downsample* the feature maps of FlowFeat and FeatUp, which effectively implements the proposed setting. Here, FlowFeat also provides a tangibly higher accuracy across all baselines.
>
> Overall, these results demonstrate that the gains come not merely from the increased resolution, but from the improved data representation as well.
>
> > **Q.3 Regarding W2: Could the authors please elaborate on the motivation behind these choices?**
>
> As discussed above, the setting (d) in Tab. 3 already trains a model with the weight set to zero (i.e. no flow reconstruction loss). This leads to a slight decrease in the downstream accuracy on VOS.
>
> > **Q.4 Are the representations directly related to the base encoder's representations in any way?**
>
> We are not sure about the question, but the FlowFeat representation comes from the DPT decoder, which has skip connections to the encoder model – this defines one relation. Moreover, FlowFeat is motion-driven, as it focuses on the foreground elements with large expected motion. By contrast, the base encoder’s representation is appearance-driven. Therefore, the two representations are different and complementary.
>
> > **Q.5 Limitations are touched upon only very briefly and spread throughout the paper rather than being consolidated in a single place. [...]**
>
> Thank you for the great suggestion. We will consolidate the limitations in one place. Note that NeurIPS will provide an additional page for the camera-ready version, and we will be happy to use the allocated space to address this concern.
>
> > **Q.5 [...] The PCA visualizations also appear to place a significant emphasis on moving objects [...]**
>
> This is an accurate observation and an expected outcome of motion parallax. However, our experimental protocol with semantic segmentation and depth estimation empirically validate that FlowFeat still preserves useful background information. We further elaborate on this point above in our response to reviewer **meu9**, to which we kindly refer.
>
>
> **Thank you for considering our response. We hope that it addresses your concerns and provides grounds for a more endorsing recommendation.**

---

> > ### Comment · Reviewer_2G2T · 2025-08-05
> >
> > I thank the authors for the clarifications and additional experiments. Some of my concerns have been resolved. I have some further questions and comments below:
> >
> > > As can be seen from our qualitative examples (c.f. Figs. 1,3,4,5), FlowFeat exhibits fine-grained boundaries, suggesting no difficulty in recovering high-frequency information.
> >
> > There seems to have been a misunderstanding as to what I was criticizing. I was not criticizing that the evals do not show that high-frequency information *can potentially* be recovered. I was criticizing that this ability is wholly dependent on the frozen feature encoder providing high-frequency information, which can not generally be expected and limits generality of the resolution improvement compared to methods like FeatUp
> >
> > > The refinement with PAMR will correspondingly "smudge" the segmentation masks, if used with the FeatUp’s feature maps, degrading the segmentation accuracy.
> >
> > It would be nice if this claim were supported by quantitative evidence.
> >
> > > This is an interesting idea. However, our experimental setting already accounts for the possibility of ‘mere’ upsampling of the feature representation in two ways.
> >
> > I still think the proposed experiment could improve the paper by isolating this property. This is, however, not an important point for the discussion, and I do not expect authors to perform this experiment during the discussion period.

---

> > > ### Author Response · Authors · 2025-08-05
> > >
> > > Thank you for reading our response. We are happy that it resolved some of the concerns and elaborate further next.
> > >
> > >
> > > > **[FlowFeat is]  wholly dependent on the frozen feature encoder providing high-frequency information [...]**
> > >
> > > We agree and we will revise our section on limitations accordingly.
> > >
> > > > **[high-frequency information in the frozen feature encoder] can not generally be expected and limits generality of the resolution improvement compared to methods like FeatUp**
> > >
> > > We agree that FlowFeat may not apply to some backbone networks. However, DINO, MAE and DINOv2 are one of the most commonly used pre-training methods in our community. Therefore, we believe that FlowFeat may still find broad (though not universal) generalisation.
> > >
> > > Regarding the generality of FeatUp, we kindly note that our experiments actually provide evidence to the contrary. Although trained on a fixed resolution (224x224), FlowFeat generalises compellingly to a higher resolution at test-time (448x448); FeatUp does not. We kindly refer to Tab. 4 in the supplemental material and the supplemental video **b_pca_feat.mp4**.
> > >
> > > > **[FeatUp++] It would be nice if this claim were supported by quantitative evidence.**
> > >
> > > Our experiments with applying PAMR to FeatUp did not yield any notable improvement – at least not to the extent we observed with FlowFeat. For DINO2-S14, PAMR improves FeatUp's mIoU only marginally (by +0.2% mIoU / +0.1% pAcc). For DINO-S16, the improvement is also modest: +0.5% / +0.4% mIoU / pAcc. In both cases, FlowFeat++ achieves a higher segmentation accuracy by a margin.
> > >
> > > | Method    | mIoU | pAcc |
> > > | -------- | ------- | ------- |
> > > | FeatUp / DINO-S16 |  41.6 |  69.5 |
> > > | FeatUp / DINO-S16 + PAMR |  42.1  | 69.9 |
> > > | FeatUp / DINO2-S14  | 58.3  | 79.1 |
> > > | FeatUp / DINO2-S14 + PAMR  |  58.5  | 79.2 |
> > >
> > >
> > > > **I still think the proposed experiment could improve the paper by isolating this property.**
> > >
> > > Thank you again for making this suggestion. We will add another experiment to clearer disentangle the contributions of feature resolution and the actual representation.

---

> > > > ### Comment · Reviewer_2G2T · 2025-08-06
> > > >
> > > > Thank you for the additional clarifications and the additional evaluation.
> > > >
> > > > Given the context of the additional evaluation, I agree that PAMR only marginally improves the FeatUp results as previously claimed.
> > > >
> > > > At this point, the rebuttal has addressed most of my concerns, and I do not see any other open problems raised by other reviewers during the review process that I consider critical. Therefore, I will increase my score and hope for acceptance.
> > > >
> > > > I hope that the authors will incorporate all suggestions and additional relevant discussions & experimental results presented during the rebuttal & discussion phase in the updated version of the paper to improve the clarity & rigor of the paper.

---

> > > > > ### Author Response · Authors · 2025-08-06
> > > > >
> > > > > We appreciate the discussion and your feedback, which we will incorporate. Thank you!

---

### Official Review · Reviewer_7P8o · 2025-07-01

**Clarity:** 2
**Significance:** 2
**Originality:** 2
**Rating:** 4
**Confidence:** 3

**Summary:**

This paper introduces FlowFeat, a novel, high-resolution feature representation for images that is learned in a self-supervised manner from video data. The core idea is to distill "motion profiles"—distributions of plausible motion—from optical flow estimates into a dense, pixel-level feature map. This is achieved by training a lightweight decoder on top of a frozen, pre-trained image encoder. The training objective forces the learned features to be able to reconstruct the optical flow between pairs of video frames via a dynamically computed linear transformation. The authors demonstrate that FlowFeat significantly enhances the performance of several state-of-the-art visual encoders on a variety of dense prediction tasks, including video object segmentation, monocular depth estimation, and semantic segmentation. The method is shown to be label-efficient, computationally inexpensive to train, and robust to inaccuracies in the underlying optical flow estimation, offering a practical step toward creating more versatile and detailed image representations.

**Questions:**

1. In Section 4.1, you state that for linear probing on DAVIS-2017, a linear classifier is trained on the ground-truth segmentation of the first frame. Given that the first frame contains only a single annotated instance for each object, how do you construct a balanced training set for the linear classifier, especially for distinguishing the object from a diverse background?

2. The ablation study in Table 3 shows that removing the L1 reconstruction loss (experiment d) leads to only a minor drop in accuracy, suggesting the gradient loss L_{∇} is doing most of the work. This is surprising. Can you explain the reason?

**Ethical Concerns:**

["NO or VERY MINOR ethics concerns only"]

**Final Justification:**

Thank you to the authors for their responses. I find the work interesting and valuable to the community. My main concern was the lack of theoretical analysis in the proposed learning-based model. However, as the authors clarified during the rebuttal phase, providing such an analysis may be non-trivial. Therefore, I am willing to raise my score. That said, I strongly recommend including a self-consistent theoretical analysis in the camera-ready version.

**Limitations:**

Yes

**Quality:**

2

**Strengths And Weaknesses:**

Strengths

1. The core idea of embedding "motion profiles" by learning to approximate optical flow with a distribution of linear transformations is both novel and well-motivated. It leverages the strong geometric and semantic cues inherent in motion without requiring expensive manual annotations.

2. The paper presents a comprehensive set of experiments across three distinct and challenging dense prediction tasks. FlowFeat consistently and significantly improves upon strong baseline encoders (MAE, DINO, DINOv2) and outperforms or performs competitively with alternative upsampling techniques like FeatUp. The improvements are particularly notable for weaker encoders like MAE.

3. The ablation studies and qualitative results demonstrate that the method is robust to the choice of the optical flow network and resilient to inaccurate or artifact-prone flow estimations, which is a critical property for real-world application.



Weakness

1. The training relies on the assumption that two different, overlapping crops from the same initial frame should be able to predict the flow in their corresponding regions of the target frame using the same linear operator A *. While this works well empirically, the justification for this core assumption could be stronger.

2. While the paper commendably points out a limitation with non-rigid objects (the snake example), a broader discussion of potential failure modes would be beneficial. For example, how does it handle scenes with very little or no motion, or scenes with extremely complex, non-linear motion patterns?

3. The authors state that state-of-the-art networks, such as transformers, generate low-resolution feature grids that are suboptimal for dense prediction tasks. How is this suboptimality quantified or evaluated in practice?

---

> ### Author Rebuttal · Authors · 2025-07-29
>
> Thank you for your time and your valuable feedback.
>
> We agree with both of your main points: the justification of the assumptions and the discussion of the limitations could be expanded upon, for which we had limited space in the submission. NeurIPS will provide an additional page for the camera-ready version, and we will be happy to use the allocated space to address your concerns, as we do below:
>
> >  **[...] the justification for this core assumption could be stronger.**
>
> The main idea behind the assumption derives from the following observation:
> the apparent motion (optical flow) is structured and spatially correlated. Therefore, the motion of two overlapping crops will be correlated as well. We model this correlation by a shared linear mapping A*, which predicts optical flow in both crops. As a result, the feature representation in both random crops will have a (partially) shared and spatially correlated structure.
>
> Note that the intuition behind the cropping is not dissimilar to previous work (e.g. DINO), where two random crops are required to share a (global) representation. Our novelty is a dynamic computation of the global representation (A*), which additionally allows for efficient learning of per pixel motion-aware representations.
>
> > **[...] how does it handle scenes with very little or no motion, or scenes with extremely complex, non-linear motion patterns?**
>
> We assume that the question relates to the training process, since FlowFeat is a *monocular* model (it takes only one image as the input).
>
> When there is no motion, there is a trivial linear mapping (zero vector), so the learning gradient will be zero, effectively discarding the training sample.
>
> State-of-the-art optical flow networks can handle relatively complex non-rigid motion, hence provide useful cues for training FlowFeat also in challenging non-rigid scenarios. Moreover, training FlowFeat is not very sensitive to errors in optical flow – thanks to the ridge regularisation in the least-squares solver (c.f. Eq. (4)) and the (robust) L1 training loss. However, we excluded compilation videos (i.e. multiple short clips stacked together) from the Kinetics dataset before training, since they break motion continuity.
>
> > **The authors state that state-of-the-art networks, such as transformers, generate low-resolution feature grids that are suboptimal for dense prediction tasks. How is this suboptimality quantified or evaluated in practice?**
>
> All our experiments with the baselines are dense tasks, which exemplify one way of quantifying and evaluating the low-resolution feature grids. This setting corresponds to our baselines in Tabs. 1 and 2 (e.g. DINO, MAE).
>
> Qualitatively, the limitations become particularly apparent in Figs. 1, 4 and 5.
>
> > **[...] how do you construct a balanced training set for the linear classifier, especially for distinguishing the object from a diverse background?**
>
> We do not finetune or re-balance the training set for the linear probe. Indeed, the classification problem for fitting the linear probe in VOS can be highly imbalanced. However, this is not an issue, because the imbalance observed in the first frame may reflect the imbalance in the rest of the video and can therefore serve as a useful bias. A heuristic-free probing strategy is more transparent and makes the results more interpretable.
>
> > **The ablation study in Table 3 shows that removing the L1 reconstruction loss (experiment d) leads to only a minor drop in accuracy [...]**
>
> This is because motion discontinuities are known to correlate well with semantic boundaries.
> Recall that we compute the gradient loss based on our lower-bound flow approximation with $A^\ast$ (c.f. Eq. (6)). Therefore, minimising the gradient loss with the solution $A^\ast$ will also lead to a lower reconstruction loss (c.f. Eq. (5)). This implies that the gradient loss jointly performs two functions: it focuses on motion discontinuities while minimising the reconstruction loss.
>
> In practice, we observed that training FlowFeat with the gradient loss alone may be unstable for some backbone models and datasets -- presumably due to the spatial sparsity of the learning gradient (the loss vanishes for pixel areas with constant motion). Therefore, we keep both loss terms for consistency across all our experiments.
>
>
> **We hope that our response resolves your concerns and look forward to a more favourable recommendation.**

---

> > ### Comment · Reviewer_7P8o · 2025-08-01
> >
> > Thanks the authors for the responses. I keep my rating unchanged

---

> > > ### Author Response · Authors · 2025-08-01
> > >
> > > Dear Reviewer,
> > >
> > > thank you for reading our response. Given the core idea’s novelty and strong empirical results, which your initial review acknowledged, we would like to understand the reasons for keeping the “borderline reject” rating. We are happy to clarify further, while we still have the possibility to do so in the open discussion phase.

---

> > > > ### Comment · Area_Chair_Yo5f · 2025-08-04
> > > >
> > > > I assume that reviewer 7P8o has asked all relevant questions and will elaborate during the closed discussion phase.

---

> ### Comment · Reviewer_7P8o · 2025-08-08
>
> While the authors noted that ‘in practice, we observed that training FlowFeat with the gradient loss alone may be unstable for some backbone models and datasets,’ a rigorous theoretical analysis would provide stronger support for this observation.

---

> > ### Author Response · Authors · 2025-08-08
> >
> > Thank you for your suggestion.
> >
> > We agree that a theoretical analysis of this observation could be interesting. However, such an analysis is known to be highly non-trivial with deep networks trained with stochastic optimisation. A plausible hypothesis for the observed instability is the high variance of the learning gradient. This would not be very surprising, since the boundary loss comes from motion discontinuities — a relatively sparse signal compared to the per-pixel reconstruction loss.
> >
> > We will incorporate this hypothesis in the revision, and will highlight the need for a more rigorous theoretical analysis as an interesting future endeavour.

---

### Official Review · Reviewer_meu9 · 2025-07-05

**Clarity:** 3
**Significance:** 3
**Originality:** 4
**Rating:** 5
**Confidence:** 4

**Summary:**

This paper proposes a new method for training an upsampling decoder for image features in a self-supervised way by taking advantage of video data. The goal is to take image features from a frozen, large-scale pre-trained backbone (encoder) and upsample them to dense features for better performance on dense tasks such as semantic segmentation and depth estimation. The work is most similar to LoftUp and FeatUp which are other feature upsampling methods with similar use cases. FlowFeat claims to be different by not requiring runtime optimization like FeatUp and also not relying on pre-trained models such as SAM to generate full resolution pseudo-GT like LoftUp. By contrast, FlowFeat claims to be fully unsupervised, fast to inference, efficient to train, and also comes with temporal consistency advantages compared to other methods.

The primary method for generating these upsampled features is by using a DPT style decoder head which is trained to predict dense "motion profiles" from a single image. This isn't quite optical flow because predicting flow from a single image is an under constrained task, thus the actual goal is to train a feature vector such that a simple linear transformation will produce the correct flow for any temporally neighboring frame. The supervision for these flow outputs comes from a pre-trained optical-flow network such as RAFT or SMURF.

**Questions:**

1. How much feature map upsampling are you able to achieve with good results using FlowFeat?
2. Do the random crops used as input to the flow model have to be a specific resolution?
3. Does the resolution of the flow network's outputs affect the DPT decoder's ability to upsample?

**Ethical Concerns:**

["NO or VERY MINOR ethics concerns only"]

**Final Justification:**

The authors adequately answered my questions, and clarified the slight limitations of their proposed method. I do not see this as a fundamental issue since the method is still a strict improvement over the baseline. I maintain my rating to accept.

**Limitations:**

Given the reliance on flow, I'd like to see the authors dive a bit more into the performance on "static background" elements in images (for example, segmentation performance on a tree in the background). This seems like it may be a limitation, but is hard to see from the qualitative examples provided. Additionally pointing out that the method only really applies to RGB web images, and not other domains is a good idea (as mentioned in the weaknesses section).

**Quality:**

4

**Strengths And Weaknesses:**

## Strengths
The main strength of FlowFeat is it's unique formulation which makes it generalizable (no per-frame optimization) and inexpensive to train. The core idea of using motion cues and optical flow as a supervisory signal for dense features is brilliant and it clearly has positive impacts of tasks requiring temporal consistency such as video object segmentation. The linear transformation & motion profile section is also well explained, and in combination with a public code release makes a compelling case for wider adoption and testing in a wide variety of tasks. One example I can think of is that I suspect the features produced by FlowFeat might make for a good drop-in replacement for those used by point tracking methods.

## Weaknesses
The main weaknesses of the paper are twofold. One is that the flow based features appear to lose background features when visualized the same way as those from FeatUp (for example). This is best seen in Figure 1 in the skateboarding example where FlowFeat seems to smush the background (sky tree etc.) into an amorphous blob. It's also visible in the videos provided in the supplemental. The second is that it's positioned as an unsupervised method, which I guess technically it is (especially if you use SMURF) but needing a good flow estimation network does limit it's applicability somewhat (for example, to other domains such as medical images where pre-trained flow methods might not work as well). Though really I consider this second weakness to be more of a nitpick.

---

> ### Author Rebuttal · Authors · 2025-07-29
>
> Thank you for your considerate, very encouraging feedback.
>
> > **"[...] flow based features appear to lose background features when visualized [...]" / "[...] dive a bit more into the performance on "static background" elements [...]"**
>
> This is an accurate observation. Due to motion parallax, FlowFeat will prioritise foreground elements, where the apparent motion is expected to have a larger magnitude than that of the background pixels. However, FlowFeat also retains useful pixel-level information about the background structure. We show this empirically in Tab. 2 and elaborate further next.
>
> Recall that NYUv2 is composed of static elements, which are semantically akin to background. On this benchmark, the improvements of FlowFeat over the baselines are consistent across all scenarios and notably surpass FeatUp. This suggests that FlowFeat not only preserves the static details (also visualised in Fig. 3, third row), but also leverages motion parallax as a useful depth cue.
>
> Our segmentation results on COCO-Stuff are similarly consistent. As suggested, we further inspected the segmentation accuracy of non-dynamic categories. Specifically, we analysed the per-category IoU for semantic classes attributable to background areas (e.g. “trees”). As we report below, FlowFeat improves on those categories across all models, though somewhat less significantly than on (potentially) dynamic objects. For reference, we also provide the accuracy for ‘person’, as a typical foreground instance.
>
> | Model | Person | Wall | Landscape | Vegetation | Ground |
> |-----------|-----------|-----------|-----------|-----------|-----------|
> | DINO S16 | 69.3 | 46.5 | 43.9 | 65.5 | 33.3 |
> | +FlowFeat-YT | **75.6** | **50.0** | **50.8** | **69.9** | **37.0** |
>
> | Model | Person | Wall | Landscape | Vegetation | Ground |
> |-----------|-----------|-----------|-----------|-----------|-----------|
> | DINO B16 | 72.9 | 51.4 | 51.0 | 70.3 | 38.9 |
> | +FlowFeat-KT | **77.8** | **52.9** | **53.1** | **71.7** | **39.6** |
>
> | Model | Person | Wall | Landscape | Vegetation | Ground |
> |-----------|-----------|-----------|-----------|-----------|-----------|
> | DINOv2 S14 | 76.9 | 57.6 | 59.2 | 71.0 | 44.6 |
> | +FlowFeat-YT | **81.7** | **59.3** | **60.5** | **73.0** | **45.7** |
>
> | Model | Person | Wall | Landscape | Vegetation | Ground |
> |-----------|-----------|-----------|-----------|-----------|-----------|
> | DINOv2 B14 | 77.0 | 59.2 | 59.6 | 70.3 | 44.9 |
> | +FlowFeat-KT | **83.0** | **61.7** | **61.3** | **72.3** | **45.1** |
>
> | Model | Person | Wall | Landscape | Vegetation | Ground |
> |-----------|-----------|-----------|-----------|-----------|-----------|
> | MAE B16 | 72.2 | 50.8 | 52.7 | 66.9 | 36.1 |
> | +FlowFeat-KT | **78.6** | **51.3** | **53.7** | **69.9** | **38.8** |
>
>
> We will add analysis of this limitation to the final revision (e.g. in the additional 10th page). Thank you for the suggestion.
>
>
> > **"[...] needing a good flow estimation network does limit it's applicability somewhat [...]"**
>
> Indeed, our approach requires a pre-trained optical flow network and video data. However, if the video sequences in another domain satisfy the assumption of brightness constancy, optical flow networks will likely still operate reliably. Consequently, a fine-tuned FlowFeat on the domain would also provide reasonable representations. However, we acknowledge that the success of FlowFeat ultimately depends on the training data available in the domain.
>
> > **Q.1 How much feature map upsampling are you able to achieve with good results using FlowFeat?**
>
> As we show in the supplemental videos and Tab. 4 (supp.), scaling up the input to FlowFeat to resolution 448x448 improves the downstream accuracy on VOS further – in contrast to FeatUp. This resolution is already close to the upper bound of DAVIS (480p), which we used in the experiments. Following your suggestion, we have now inspected FlowFeat for the input resolution of 1080p. Since we cannot share any visual data, we report our observations. The PCA visualisation exhibits sharp boundaries, aligned with the image gradient and (potential) motion discontinuities. We observed no trace of artefacts akin to those emerging in FeatUp (c.f. supp. videos). Please note that evaluating transformers at resolution 1080p significantly increases their inference time and the memory footprint, thus we could not timely evaluate the models on benchmarks.
>
>
> > **Q.2 Do the random crops used as input to the flow model have to be a specific resolution?**
>
> We feed frame pairs at resolution 224x224 to the flow network. Then, we sample the random crops from the produced flow map to align them with the input and output of the FlowFeat network. Therefore, the target flow maps used to train FlowFeat are bilinearly upsampled by a factor from the range (1, 2.5]. We did not find the training to be sensitive to moderate deviations from this range.
>
> > **Q.3 Does the resolution of the flow network's outputs affect the DPT decoder's ability to upsample?**
>
> A low-resolution optical flow would conceivably result in smooth boundaries in the feature map of FlowFeat. Conversely, a high-resolution optical flow would increase the boundary sharpness, while also increasing the computational costs. We found a good compromise by running optical flow at the training resolution of FlowFeat (224x224) and upsampling the flow map to fit the input crops with bilinear interpolation, as described previously.
>
> > **[...] the method only really applies to RGB web images, and not other domains [...]**
>
> We will point out the current application scope of FlowFeat to RGB images from the Internet – thank you for suggesting it.

---

> > ### Comment · Reviewer_meu9 · 2025-08-07
> >
> > Thanks to the authors for the detailed rebuttal & for providing detailed segmentation performance. As expected, FlowFeat provides more improvement on categories that could potentially move. This makes sense, but doesn't appear to be a critical flaw of the method, as it is still a strict improvement over the baseline. My other questions have also been adequately answered and thus I maintain my rating for acceptance.

---

> > > ### Author Response · Authors · 2025-08-07
> > >
> > > Thank you for acknowledging our response and for your feedback.

---

### Decision · Program_Chairs · 2025-09-17

**Decision:**

Accept (spotlight)

**Comment:**

The manuscript addresses the problem of increasing the resolution of feature representations. Two contributions are claimed: the method, an effective self-supervised training framework that exploits flow networks and large video datasets, and an evaluation of the obtained feature representation (FlowFeat). Both parts are considered as strength by the reviewers and the AC. Main points of criticism are on evaluation details and discussion of short comings and edge cases. The weaknesses are of minor nature, while the strengths are dominating, thus the recommendation to accept the paper. As the topic of increased feature representation resolution is of broad relevance, it is suggested to present the work as spotlight, but even an oral would work.
All reviewers were rather positive from the beginning and after rebuttal and discussion, two suggest acceptance, and two BA. The main weaknesses addressed in the rebuttal were about edge cases and additional reflections. Several reviewers raised their assessment, but also pointed out to include the additional theoretical (7P8o) and intuitive (zphZ) reflections in the final version.